# Percutaneous Drainage vs. Surgery as Definitive Treatment for Anastomotic Leak after Intestinal Resection in Patients with Crohn’s Disease

**DOI:** 10.3390/jcm12041392

**Published:** 2023-02-09

**Authors:** Angela Belvedere, Gerti Dajti, Cristina Larotonda, Laura Angelicchio, Fernando Rizzello, Paolo Gionchetti, Gilberto Poggioli, Matteo Rottoli

**Affiliations:** 1Surgery of the Alimentary Tract, IRCCS Azienda Ospedaliero Universitaria di Bologna, 40138 Bologna, Italy; 2Department of Medical and Surgical Sciences, Alma Mater Studiorum University of Bologna, 40128 Bologna, Italy; 3IBD Unit, IRCCS Azienda Ospedaliero Universitaria di Bologna, 40138 Bologna, Italy

**Keywords:** anastomotic leak, Crohn’s disease, percutaneous drainage, image-guide drainage, perianastomotic collection, intra-abdominal abscess, bowel resection, ileostomy

## Abstract

Background: Anastomotic leak (AL) remains one of the most relevant complications after intestinal resection for Crohn’s disease (CD). While surgery has always been considered the standard treatment for perianastomotic collection, percutaneous drainage (PD) has been proposed as a potential alternative. Methods: Retrospective study in consecutive patients treated with either PD or surgery for AL after intestinal resection for CD between 2004 and 2022. AL was defined as a perianastomotic fluid collection confirmed by radiological findings. Patients with generalized peritonitis or clinical instability were excluded. Primary aim: To compare the success rate of PD vs. surgery. Secondary aims: To compare the outcomes at 90 days after the procedures; to identify the variables associated with the indication for PD. Results: A total of 47 patients were included, of which 25 (53%) underwent PD and 22 (47%) surgery. The success rate was 84% in the PD and 95% in the surgery group (*p* = 0.20). There were no significant differences between the PD and surgery group in postoperative medical and surgical complications, discharge, readmission or reoperation rates at 90 days. PD was more likely to be performed in patients with later diagnosis of AL (OR 1.25, 95% CI 1.03–1.53, *p* = 0.027), undergoing ileo-colic anastomosis alone (OR 3.72, 95% CI 2.29–12.45, *p* = 0.034) and treated after 2016 (OR 6.36, 95% CI 1.04–39.03, *p* = 0.046). Conclusion: The present study suggests that PD is a safe and effective procedure to treat anastomotic leak and perianastomotic collection in CD patients. PD should be indicated in all eligible patients as an effective alternative to surgery.

## 1. Introduction

The risk of developing postoperative complications is higher after bowel resection for Crohn’s disease (CD) than after surgery for other benign and malignant colorectal conditions [1,2]. Anastomotic leak remains one of the most feared complications, occurring in 6.4–14% of CD patients undergoing bowel resection and anastomosis [3,4].

While reoperation has been the standard treatment for anastomotic leak for decades, percutaneous drainage (PD) has shown positive results as an alternative to surgery. In particular, avoiding surgery is associated with shorter hospital stay, lower morbidity, as well as a lower risk of intestinal failure given the recurrent nature of CD in the long-term [5,6,7,8].

There are limited data regarding the outcomes of PD as a treatment for anastomotic leak in the specific population of patients affected by CD who underwent bowel resection [9].

The aim of this study was, therefore, to analyze the effectiveness and safety of PD by comparing the success rates and post-procedure outcomes of PD and surgery as a primary treatment for anastomotic leak in CD patients.

## 2. Materials and Methods

The present study was a retrospective single-center study including all consecutive patients aged 18 years and older with CD who underwent bowel resection and anastomosis complicated by anastomotic leak, between November 2004 and June 2022. A flow diagram showing eligible, excluded, and included patients in the final cohort is presented in Figure 1. The study was approved by the Institutional Ethical Committee.

The study period was chosen in order to include only the patients who could have potentially undergone biological therapy before surgery [9,10]. The anastomotic leak was defined as a collection or an abscess adjacent to the anastomosis, clinically symptomatic, confirmed by radiological findings (ultrasound or CT scan), within 30 days after surgery [11].

Exclusion criteria: the presence of intra-abdominal collections not related to anastomotic leak; non-symptomatic perianastomotic collection; small, symptomatic perianastomotic collections resolved with antibiotic therapy alone; generalized ascites; generalized fecal peritonitis; anastomotic leak requiring emergency surgery due to clinical patient instability.

Patients underwent either PD or surgery as a primary treatment for the anastomotic leak-associated abscess. The decision regarding which primary treatment was indicated was made on a case-by-case basis, subject to clinical evidence and surgeons’ experience. A wide-spectrum beta-lactam antibiotic therapy was commenced at the time of diagnosis, which could have later been modified according to the antibiogram.

The success of the primary treatment was defined as the clinical and radiological resolution of the complication at 90 days after the treatment. The failure of the treatment was defined as the persistence or relapse of the complication requiring a subsequent invasive procedure (either surgical or percutaneous).

The dataset included the demographic characteristics, the medical therapy, the perioperative details and the postoperative outcomes at 90 days after the date of treatment, and was retrieved from an institutional database using REDCap [12]. The primary aim of the study was to compare the success rate of PD and surgery as primary treatments for anastomotic leak.

The secondary aim was to compare the rate of medical and surgical complications at 90 days after the procedure.

Statistical analysis was performed using STATA Software (version 17.0). Parametric and non-parametric tests were used accordingly to compare groups. Continuous and categorical variables are described as medians with the interquartile range (IQR) and percentages, respectively. A multivariate regression model was built in a backward stepwise fashion including only variables with a p value of at most 0.100 in the univariate analysis. The threshold for statistical significance was set at 0.05.

## 3. Results

A total of 47 CD patients were identified who developed anastomotic leak. Fifty-three percent (25/47) underwent PD and 47% (22/47) underwent surgery as primary treatment for anastomotic leak. A CT-guided rather than US-guided procedure was the preferred choice in the case of PD (23 vs. 2 cases, 92% vs. 8%). Table 1 reports the univariate comparison between the two groups.

Ileo-cecal resection was the most frequent type of surgery in 25 patients (53%), followed by ileo-cecal associated with sigmoid resection and double ileo-colic and colorectal anastomosis in eight patients (17%) and colectomy with ileo-rectal anastomosis in six patients (13%). Ileo-cecal resection and colectomy were approached laparoscopically in 42 cases (90%). Side-to-side extracorporeal and handsewn side-to-end were the most frequent types of construction in cases of ileo-colic and ileo-rectal anastomosis, respectively.

The PD treatment was significantly associated with longer time to diagnosis of the leak from primary surgery (9 vs. 5 days, *p* = 0.001) and a greater abscess diameter on computed tomography (60 vs. 42.5 mm, *p* = 0.007), as compared to the surgery group.

The success rates of PD and surgery were 84% and 95%, respectively (*p* = 0.204). The median time to resolution was 14 days in PD and 12.5 days in the surgery group (*p* = 0.915). The surgical procedures performed in the surgery group included: resection of the anastomosis with terminal ileostomy in 17 cases (77.3%), resection and reconstruction of the anastomosis with protective loop-ileostomy in three cases (13.6%) and without protective stoma in two cases (9.1%).

The rates of medical (12% vs. 18%, *p* = 0.55) and surgical (24% vs. 36%, *p* = 0.62) complications were comparable between the PD and surgery groups.

Table 2 shows the univariate and multivariable analysis of the variables associated with PD.

PD was significantly associated with a late diagnosis of the anastomotic leak (OR 1.25, 95% CI 1.03–1.53, *p* = 0.027), with a single ileo-colic anastomosis (OR 3.72, 95% CI 2.29–12.45, *p* = 0.034) and with the study period after 2016 (OR 6.36, 95% CI 1.04–39.03, *p* = 0.046).

## 4. Discussion

The management of intra-abdominal collection due to anastomotic leak is yet to be standardized. According to the guidelines of the International Study Group of Rectal Cancer (ISREC), the management of an intraperitoneal Grade B leak (type of anastomotic leak requiring active therapeutic intervention short of a laparotomy) should be dictated by the patient’s clinical picture. PD should also be considered for larger abscesses (>3 cm), and in case of failure, surgical drainage should be the standard treatment [11,13].

Since CD surgery is associated with a higher risk of anastomotic complications, rationalizing the use of PD as a possible unique treatment is of primary importance. The analysis confirmed that PD was associated with a high chance (84%) of resolution of the collection associated with the anastomotic leak, with no signs of recurrence at 90 days after the procedure. The rest of the patients (four, 16%) required a reoperation and had an outcome that was comparable to that of the 22 patients who underwent a redo surgery immediately after the diagnosis of the anastomotic leak [14]. The choice of surgery came at a high price, since all patients required resection of the anastomosis (followed by a redo anastomosis only in 22.7% of the cases), and the great majority (90.9%) were discharged with an ileostomy [15]. Therefore, the potential perioperative risk of complications in these patients extends also to the subsequent operation of ileostomy reversal, and should include the well-known morbidity associated with the presence of a stoma [16].

Not surprisingly, patients who underwent a reoperation had a higher chance of developing postoperative medical (18% vs. 12%) and surgical (36% vs. 24%) complications, although the difference was not statistically significant [17].

The choice of PD was more likely in case of a leak of an ileocolic anastomosis (as compared to colorectal, ileo-ileal, ileorectal or multiple anastomoses) (OR 6.34, 95% CI 1.15–35.04, *p* = 0.034), a delayed diagnosis after surgery (OR 1.25, 95% CI 1.03–1.53, *p* = 0.046) and after 2016 (OR 1.25, 95% CI 1.03–1.53, *p* = 0.027). While the latter evidence was obviously associated with the improvement in the expertise of the interventional radiology team and the increasing evidence regarding the role of PD as a valid alternative to surgery in case of intra-abdominal abscesses [18,19], the other variables reflected the cases in which the anastomotic leak was associated with a less severe clinical picture, thus making the indication for PD more straightforward. As previously stated, though, patients who had a diffuse peritonitis or presented with a deteriorated clinical picture requiring emergency surgery were excluded from the present study, including those who could have potentially undergone either treatment (surgery or PD).

The safety and effectiveness of image-guided PD as a treatment for anastomotic leak following colorectal surgery have been previously analyzed [20]. Specifically in CD patients, the percutaneous procedure is indicated in the guidelines in case of a preoperative diagnosis of an abscess, in order to postpone surgery and reduce the risk of extensive resection and intraoperative complications [21,22,23,24]. However [24], there has been a lack of reports in the literature regarding the outcomes of the procedure as a primary treatment for anastomotic leak after surgery for CD [25,26].

The present study has a few limitations. The first one regards the retrospective nature of the study. In addition, we acknowledge that the current design was underpowered to detect significant differences due to the limited sample size. On the other hand, this was the first study comparing two groups of CD patients with similar clinical and surgical features, who experienced a postoperative anastomotic leak and were treated with either PD or surgery, in a single referral center for IBD. The results suggested that PD must be considered as a primary treatment in case of anastomotic leak associated with a perianastomotic abscess also in patients affected by CD. An algorithm of treatment should be developed in order to offer these patients the best short- and long-term outcomes.

## Figures and Tables

**Figure 1 jcm-12-01392-f001:**
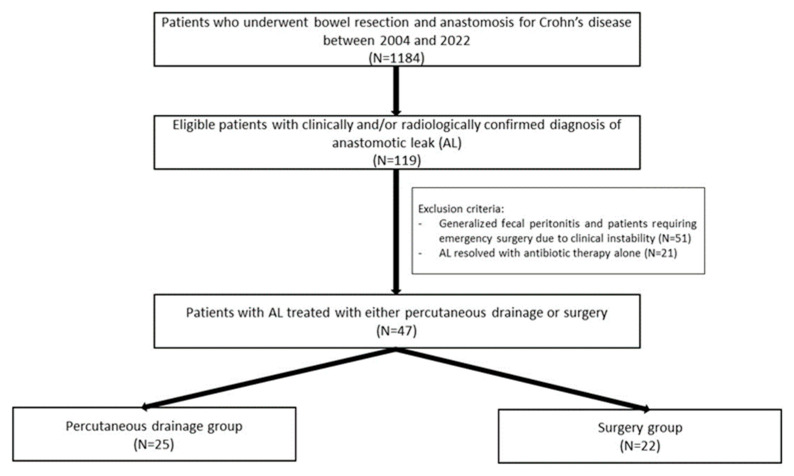
Flow diagram of included patients in the study.

**Table 1 jcm-12-01392-t001:** Patients’ characteristics, surgery features and follow-up.

	Overall (*n* = 47)	Percutaneous Drainage (PD) (*n* = 25)	Surgery (*n* = 22)	*p* *
Patient and preoperative features				
Age (years)	45 (33–56)	48 (38–53)	37.5 (31–56)	0.2
Male gender	32 (70%)	18 (76%)	14 (64%)	0.4
BMI (kg/m^2^)	22.0 (19.5–24.6)	22.6 (21.3–24.6)	20.8 (19.0–24.5)	0.3
Steroids at surgery	20 (45%)	11 (44%)	9 (40%)	0.9
History of biologic therapy	19 (40%)	12 (48%)	7 (32%)	0.3
Disease behavior				
stenotic	33 (72%)	18 (76%)	15 (68%)	0.6
penetrating	23 (49%)	13 (52%)	10 (45%)	0.7
Abscess at surgery	8 (17%)	4 (16%)	4 (18%)	0.8
Previous abdominal surgery	21 (45%)	14 (56%)	7 (32%)	0.1
Serum albumin (g/dL)	3.9 (3.6–4.3)	3.8 (3.5–4.1)	4.1 (3.6–4.4)	0.040
Hemoglobin (g/dL)	13.0 (12.1–14.3)	13.1 (11.9–14.4)	12.9 (12.6–14.1)	0.9
Diagnosis and surgical features				
POD of leak	7 (5–12)	9 (7–16)	5 (4–6)	0.001
Abscess diameter (mm)	50 (40–63)	60 (47–80)	42.5 (35–50)	0.007
WBC (10^3^/mm^3^)	12.5 (9.2–14.5)	11.4 (9.2–14.3)	13 (8.8–16.2)	0.4
Serum CRP (mg/dL)	13.4 (8.9–18.7)	12.3 (7.8–18.6)	15.9 (9.9–22.6)	0.3
Procalcitonin (ng/mL)	0.6 (0.1–2.8)	0.1 (0.1–6.9)	0.9 (0.2–2.8)	0.5
Year of intervention				0.1
2004–2015	28 (60%)	12 (43%)	16 (57%)	
2016–2022	19 (40%)	13 (68%)	6 (32%)	
Type of anastomosis				
ileo-colic	25 (53%)	17 (68%)	8 (36%)	0.030
ileo-colic and colo-rectal	8 (17%)	3 (12%)	5 (23%)	0.5
ileo-colic and ileo-ileal	4 (8.5%)	3 (12%)	1 (4%)	0.3
ileo-ileal	4 (8.5%)	1 (4%)	3 (14%)	0.6
ileo-rectal	6 (13%)	1 (4%)	5 (23%)	0.1
Follow-up				
Success rate of primary treatment	42 (89%)	21 (84%)	21 (95%)	0.2
Resolution time (days)	13 (8–26)	14 (6–38)	13 (10–22)	0.9
Follow-up (months)	19 (12–58)	17 (11–56)	21 (12–69)	0.7
Discharge at 90 days	46 (98%)	25 (100%)	21 (95%)	0.3
Re-admission at 90 days	2 (4%)	2 (8%)	0 (0%)	0.2
Re-intervention at 90 days	6 (13%)	4 (16%)	2 (9%)	0.5
Medical complications at 90 days	7 (15%)	3 (12%)	4 (18%)	0.6
Pneumonia	3 (6%)	1 (4%)	2 (9%)	0.5
Septic	2 (4%)	2 (8%)	0 (0%)	0.2
Other	2 (4%)	0 (0%)	2 (9%)	0.1
Surgical complications at 90 days	10 (21%)	6 (24%)	4 (36%)	0.6
Wound infection	3 (6%)	2 (8%)	1 (5%)	0.6
IASC	6 (13%)	4 (16%)	2 (9%)	0.5
Other	1 (2%)	0 (0%)	1 (5%)	0.3

Variables were reported as median (interquartile range) or *n* (%). BMI: Body mass index; POD: postoperative day; WBC: white blood cells; CRP: C reactive protein; IASC: intra-abdominal septic complications. * Kruskal–Wallis test for non-parametric continuous variables; chi-square test for categorical values.

**Table 2 jcm-12-01392-t002:** Univariate and multivariate analysis of variables associated with percutaneous drainage.

	Univariate Analysis		Multivariate Analysis	
	OR (CI 95%)	*p*	OR (CI 95%)	*p*
Age	1.02 (0.98–1.06)	0.4		
Male gender	0.55 (0.16–1.96)	0.4		
BMI (kg/m^2^)	1.03 (0.90–1.17)	0.7		
Steroids	0.94 (0.30–2.99)	0.9		
Biologics	1.98 (0.60–6.51)	0.3		
Stenotic disease	1.48 (0.41–5.34)	0.5		
Penetrating disease	1.30 (0.41–1.40)	0.7		
Abscess at surgery	0.86 (0.19–3.93)	0.8		
Previous abdominal surgery	2.73 (0.83–9.01)	0.100		
Serum albumin (g/dL)	0.88 (0.77–1.01)	0.051		
Hemoglobin (g/dL)	0.91 (0.69–1.20)	0.5		
POD leak	1.24 (1.03–1.49)	0.020	1.25 (1.03–1.53)	0.027
Abscess diameter (mm)	1.05 (1.01–1.09)	0.021		
WBC (10^3^/mm^3^)	0.92 (0.80–1.06)	0.2		
Serum CRP (mg/dL)	0.95 (0.86–1.04)	0.3		
Procalcitonin (ng/dl)	1.21 (0.59–2.45)	0.6		
Year of intervention (2016–2022 vs. before)	2.89 (0.85–9.82)	0.089	6.36 (1.04–39.03)	0.046
Ileo-colic anastomosis alone (vs. other)	3.72 (2.29–12.45)	0.033	6.34 (1.15–35.04)	0.034

OR: Odds Ratio; CI: Confidence Interval; BMI: Body Mass Index; POD: Postoperative Day; WBC: White Blood Cells; CRP: C Reactive Protein.

## Data Availability

Data supporting the findings of the study are available from the corresponding author (MR) on request according to national and international regarding privacy and data protection.

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
