# Peer review of "Percutaneous Drainage vs. Surgery as Definitive Treatment for Anastomotic Leak after Intestinal Resection in Patients with Crohn’s Disease"

_jcm, 2023, doi:10.3390/jcm12041392_

Round 1
Reviewer 1 Report
The paper "Percutaneous drainage vs surgery as definitive treatment for anastomotic leak after intestinal resection in patients with Crohn’s disease" is well designed.and written concerning a topic rarely reported in the current literature and could be very useful to change hystorical classical surgical strategies to manage common complications such as leaks and abscesses in Crohn patients operated on.
In table 1, section follow-up, line 2 , resolution time, should be added (days)
Author Response
Dear reviewers, please find enclosed the answers to your comments and requests. All of them have been acknowledged and the manuscript has been amended accordingly. Once again, we thank you for the opportunity to improve the manuscript
RESPONSE TO REVIEWER 1
Point 1: The paper "Percutaneous drainage vs surgery as definitive treatment for anastomotic leak after intestinal resection in patients with Crohn’s disease" is well designed and written concerning a topic rarely reported in the current literature and could be very useful to change historical classical surgical strategies to manage common complications such as leaks and abscesses in Crohn patients operated on.
In table 1, section follow-up, line 2, resolution time, should be added (days)
Response: Thank you for your comment. The resolution time in days was added in the manuscript.
Reviewer 2 Report
This is retrospective single-centre study comparing patient with anastomotic leak (AL) after bowel resection for Crohn’s Disease (CD) treated with surgery or percutaneous drainage (PD). The success rate between the two group did not differ significantly. Time to AL diagnosis and year of intervention (>2016) were independently associated with PD.
There are some methodological and intrinsic issues that need to be addressed.
1) The importance of avoiding further surgery and bowel resection in patients with CD and AL should be emphasized in the introduction.
2) Did you excluded patients who had protective stoma during primary surgery?
3) A flow diagram for patient inclusion/exclusion with reasons is advisable.
4) When AL occurred, have you always found a well-defined accessible intra-abdominal collection? What about multiple abscesses?
5) Please, specify the antibiotic treatment and whether there were cases of AL resolved with antibiotic therapy alone.
6) A brief description of the surgical technique for anastomotic construction should be added (e.g. side-to-side, end-to-end, handsewn, mechanical, intra/extracorporeal if a minimally invasive approach was used).
7) Does the time to resolution in table 1 correspond to clinical and/or radiological resolution?
8) As study period after 2016 is significantly associated with PD, please specify why did you set this cut-off.
9) Why the variable “abscess diameter” was not tested in the multivariate analysis?
10) Reference 20 refers to preoperative PD of intra-abdominal abscesses, not to AL treatment. Furthermore, regarding guidelines on preoperative PD for abscesses in CD patients (pg 5 lines 159-162), you could refer to Statement 3.1. ECCO CD Treatment GL (2019) (doi:10.1093/ecco-jcc/jjz187).
11) No firm conclusions can be drawn from this retrospective study with a small sample size (25 patients). Several surgeon- and patient-related factors could influence the choice to perform PD with a not negligible impact on outcomes. However, it can be argued that, in the present study, PD was as safe and effective as surgery when performed approximately more than 7 days after primary surgery and for collections > 50 mm in diameter.
12) There is no mention to ethics/ethical approval.
Author Response
Dear reviewers, please find enclosed the answers to your comments and requests. All of them have been acknowledged and the manuscript has been amended accordingly. Once again, we thank you for the opportunity to improve the manuscript.
RESPONSE TO REVIEWER 2
Point 1: The importance of avoiding further surgery and bowel resection in patients with CD and AL should be emphasized in the introduction.
Response: Thank you for the suggestion. The matter has been addressed in the introduction section according to your comment.
Point 2: Did you excluded patients who had protective stoma during primary surgery?
Response: Thank you for your comment. Patients who underwent protective stoma during primary surgery were not excluded, in order to represent the homogeneous population of patients undergoing surgery for Crohn’s disease. In particular, only one patient underwent ileocecal rection with protective ileostomy during the primary surgery.
Point 3: A flow diagram for patient inclusion/exclusion with reasons is advisable.
Response: Thank you for your suggestion. A flow diagram was included as a Figure in the manuscript.
Point 4: When AL occurred, have you always found a well-defined accessible intra-abdominal collection? What about multiple abscesses?
Response: Thank you for your comment. As specified in the methods section, all the patients who were included in the study had a peri-anastomotic collection or abscess which was amenable to be drained percutaneously. All cases with other intra-abdominal collections distant from the anastomotic site were excluded from the study.
Point 5: Please, specify the antibiotic treatment and whether there were cases of AL resolved with antibiotic therapy alone.
Response: Thank you for your question. All patients underwent wide-spectrum beta-lactam antibiotic therapy at the time of the diagnosis, which could have been modified subsequently according to the antibiogram. Given the purpose of the study, cases of small, symptomatic perianastomotic collections which resolved with antibiotic therapy alone were not included in the study. This issue has been addressed in the Materials and Methods section.
Point 6: A brief description of the surgical technique for anastomotic construction should be added (e.g., side-to-side, end-to-end, handsewn, mechanical, intra/extracorporeal if a minimally invasive approach was used).
Response: Thank you for your comment. The description of the different techniques has been reported in the Results section.
Point 7: Does the time to resolution in table 1 correspond to clinical and/or radiological resolution?
Response: Thank you for your question. The time to resolution corresponds to the clinical and radiological resolution.
Point 8: As study period after 2016 is significantly associated with PD, please specify why did you set this cut-off.
Response: Thank you for your question. The year 2016 corresponds to the median of the study period. The point was to show that the second part of the study period, therefore the most recent, was associated with the increase of the interventional radiology procedures, due to the increased expertise in our centre.
Point 9: Why the variable “abscess diameter” was not tested in the multivariate analysis?
Response: Thank you for your question. The abscess diameter was actually included during the backward stepwise multivariate analyses, but only those significantly associated with PD in the final model were shown in the table.
Point 10: Reference 20 refers to preoperative PD of intra-abdominal abscesses, not to AL treatment. Furthermore, regarding guidelines on preoperative PD for abscesses in CD patients (pg 5 lines 159-162), you could refer to Statement 3.1. ECCO CD Treatment GL (2019) (doi:10.1093/ecco-jcc/jjz187).
Response: Thank you for your comment. The manuscript and bibliography have been reviewed accordingly.
Point 11: No firm conclusions can be drawn from this retrospective study with a small sample size (25 patients). Several surgeon- and patient-related factors could influence the choice to perform PD with a not negligible impact on outcomes. However, it can be argued that, in the present study, PD was as safe and effective as surgery when performed approximately more than 7 days after primary surgery and for collections > 50 mm in diameter.
Response: Thank you for your comment, with which we agree. The limits of the study have been acknowledged in the Discussion section, as well as the need for further multicentre studies.
Point 12: There is no mention to ethics/ethical approval.
Response: Thank you for your comment. The ethical approval was reported during the submission process. A sentence was also added to the Methods section according to your comment.
Reviewer 3 Report
Belvedere et al. have presented a very interesting and important retrospective study of percutaneous drainage (PD) versus surgery for the management of anastomotic leak in Crohn's disease patients, between 2004 and 2022. A total of 47 patients were included: 25 patients in PD group and 27 patients in surgery group.
Please see minor comments below.
1. Abstract is fairly lengthy - approximately 400 words. I would aim to reduce this closer to the recommended JCM author guidelines of 200 words. "Table 2" can be removed on line 33.
2. Any data on antibiotic therapy and antibiotic duration in the groups? Similarly, any data on culture results in the groups?
Otherwise this should be mentioned in the limitations section.
3. Any data specifically on whether the PD was performed with US guidance or required CT guidance?
4. When was the CRP and white cell count recorded? The CRP count is fairly low in both groups.
5. CD as an abbreviation for Crohn's disease should be used consistently if preferred by authors.
6. Please correct keywords - "image-guided drainage" and "intra-abdominal". And please correct "intra-abdominal" throughout text where necessary.
7. Can you expand on and propose a possible treatment algorithm/practical-based recommendations based on your study findings in the discussion?
Author Response
Dear reviewers, please find enclosed the answers to your comments and requests. All of them have been acknowledged and the manuscript has been amended accordingly. Once again, we thank you for the opportunity to improve the manuscript
RESPONSE TO REVIEWER 3
Point 1: Abstract is fairly lengthy - approximately 400 words. I would aim to reduce this closer to the recommended JCM author guidelines of 200 words. "Table 2" can be removed on line 33.
Response: Thank you for your comment. The length of the abstract was reduced accordingly.
Point 2: Any data on antibiotic therapy and antibiotic duration in the groups? Similarly, any data on culture results in the groups? Otherwise, this should be mentioned in the limitations section.
Response: Thank you for your question. The details regarding the antibiotic therapy were included in the Materials and Methods section.
Point 3: Any data specifically on whether the PD was performed with US guidance or required CT guidance?
Response: Thank you for your question. The CT-guided procedure was the preferred choice most of the times. In fact, only two patients underwent a US-guided procedure. These details were added to the Results section.
Point 4: When was the CRP and white cell count recorded? The CRP count is fairly low in both groups.
Response: Thank you for your question. The reported CRP and WBC were always recorded at the time of diagnosis.
Point 5: CD as an abbreviation for Crohn's disease should be used consistently if preferred by authors.
Point 6: Please correct keywords - "image-guided drainage" and "intra-abdominal". And please correct "intra-abdominal" throughout text where necessary.
Response (to point 5 and 6): Thank you for your comment. The manuscript has been amended accordingly.
Point 7: Can you expand on and propose a possible treatment algorithm/practical-based recommendations based on your study findings in the discussion?
Response: Thank you for your suggestion. As explained in the results section, we acknowledged the limits of our study, especially those concerning the limited number of patients. Therefore, we believe that any algorithm proposal would be an overstatement, taking this into account. Our study concluded that the procedure is safe, and it should be proposed as a first-step line of treatment in case of anastomotic leak associated with perianastomotic collection after surgery for Crohn’s disease. Future, larger multicentre study will be necessary to design a strong algorithm of treatment.
Round 2
Reviewer 2 Report
All issues have been addressed.
Author Response
Dear reviewer thank you for the opportunity to improve the manuscript.